# KAN VERSUS MLP
# ON IRREGULAR OR NOISY FUNCTIONS

## ABSTRACT

In this paper, we compare the performance of Kolmogorov-Arnold Networks (KAN) and Multi-Layer Perceptron (MLP) networks on irregular or noisy functions. We control the number of parameters and the size of the training samples to ensure a fair comparison. For clarity, we categorize the functions into six types: regular functions, continuous functions with local non-differentiable points, functions with jump discontinuities, functions with singularities, functions with coherent oscillations, and noisy functions. These features are typically not available as prior knowledge in real applications; therefore, we do not specifically select the corresponding network structure for each function. Our experimental results indicate that KAN does not always perform best. Furthermore, increasing the size of training samples can improve performance to some extent. When noise is added to functions, the irregular features are often obscured by the noise, making it challenging for both MLP and KAN to extract these features effectively. We hope these experiments provide valuable insights for future neural network research and encourage further investigations to overcome these challenges.

## 1 INTRODUCTION

Since its launch, Kolmogorov-Arnold networks (KAN)(Liu et al., 2024b) has garnered significant attention. These networks utilize the Kolmogorov-Arnold representation theorem, which posits that any multivariate continuous function can be expressed as a combination of continuous univariate functions and addition. Unlike conventional Multi-Layer Perceptron (MLP) networks, KANs incorporate learnable activation functions. According to (Liu et al., 2024b), this feature provides KANs with enhanced interpretability and accuracy over MLPs.

Noting the release of KAN 2.0 (Liu et al., 2024a), we will conduct all experiments using the latest pykan version (v0.2.6) to leverage its optimizations. KAN 2.0 introduced multiplication nodes, which significantly enhance the fitting of multivariate functions, especially those involving direct multiplication or division of independent variables. However, the improvement is minimal for the functions used in this paper, so lower versions of Pykan are also acceptable.

Numerous investigations into KAN applications have rapidly surfaced, covering areas such as smart energy grid optimization(Wang et al., 2024)(Tang et al., 2024), chemistry data analysis(Wang et al., 2024)(Li et al., 2024b), image classification(Cheon, 2024)(Teymoor Seydi, 2024)(Igali & Shamoi, 2024), deep function learning(Zhang, 2024), quantum architecture search(Kundu et al., 2024), medical image analysis and processing(Li et al., 2024a)(Chen et al., 2024), disease risk predictions(Dong, 2024), graph learning tasks(Kiamari et al., 2024)(Li, 2024)(Ghaith Altarabichi, 2024), asset pricing models(Wang & Singh, 2024), 3D object detection (in autonomous driving)(Lai et al., 2024), sentiment analysis(Lawan et al., 2024), and deep kernel learning(Zinage et al., 2024).

On the contrary, a growing body of research has highlighted the imperfections of KANs compared to MLPs. For example, (Zhang, 2024) and (Shen et al., 2024) noted the vulnerability of KANs to noise, indicating that even minor noise can lead to a significant rise in test loss. Additionally, (Tran et al., 2024) claimed that KANs do not outperform MLPs in highly complex datasets and require considerably more hardware resources. Furthermore, (Yu et al., 2024) noted that MLPs generally have higher accuracy than KANs across various standard machine learning tasks, with the exception of tasks involving symbolic formula representation.

Moreover, the performance of networks may be influenced by the regularity of functions, prompting this study to compare the performance of KAN and MLP, across distinct types of functions with varying degrees of regularity (or noisy functions).

In this investigation, we assess the performance of MLP and KAN in modeling irregular or noisy functions. To ensure fairness, we control the number of parameters and the amount of training data. Moreover, we investigate the influence of different optimizers on the accuracy of fitting specific functions. This research continues directly and naturally from our recent study on the efficacy of KANs in fitting noisy functions (Shen et al., 2024).

The structure of this paper is organized as follows: Section 2 provides an introduction to the Kolmogorov-Arnold Theorem and KANs, discussing their benefits and limitations, and enumerates the six types of functions. Section 3 evaluates the performance of MLP and KAN in approximating regular and irregular functions. Section 4 introduces noise to the previously utilized functions and continues the comparison between MLP and KAN. Finally, Section 5 summarizes the findings of our experiments.

## 2 KOMOGOROV-ARNOLD THEOREM AND KANS

The Kolmogorov-Arnold theorem pertains to expressing multivariable continuous functions. According to the theorem, any continuous function involving multiple variables can be expressed as a combination of continuous single-variable functions and addition (Kolmogorov, 1956) (Kolmogorov, 1957) (Arnold, 1957). Formally, it can be stated as:

**Theorem 1.** *[Kolmogorov-Arnold Theorem] Let* $f : [0, 1]^n \to \mathbb{R}$ *be any multivariate continuous function, there exist continuous univariate functions* $\phi_i$ *and* $\psi_{ij}$ *such that:*

$$f(x_1, x_2, \ldots, x_n) = \sum_{i=1}^{2n+1} \phi_i \left( \sum_{j=1}^{n} \psi_{ij}(x_j) \right). \tag{1}$$

Leveraging the Kolmogorov-Arnold theorem, KANs introduce a novel neural network architecture. Unlike traditional Multi-Layer Perceptrons (MLPs) which use fixed activation functions, KANs employ learnable activation functions. This methodology is theoretically advantageous in enhancing the adaptability of KANs across different datasets and applications.

Unfortunately, Theorem 1 was originally proven non-constructively, lacking a constructive proof initially. In 2009, (Braun & Griebel, 2009) presented a constructive proof for this theorem. Nevertheless, challenges emerge when handling functions exhibiting irregular patterns, which mathematical analysis typically categorizes into at least five distinct types. Table 1 outlines these types along with detailed examples.

## 3 COMPARISON ON IRREGULAR FUNCTIONS

Here, we offer some explanations for the initial five types mentioned in Table 1. We compare these functions using multiple sets of KAN and MLP networks that have similar parameter counts. The number of parameters for each network are presented in Table 2. It is worth noting that the features of these functions are typically not available as prior knowledge in real applications, so we do not specifically select the corresponding network structure for each function. We implement the L-BFGS optimizer for these functions as it shows better performance in small-scale training. For functions with singularities or coherent oscillations, which might need more training samples and iterations, we also investigate the Adam optimizer's capability.

### 3.1 REGULAR FUNCTIONS

First, consider the functions exhibiting strong regularity. Such functions are continuous and differentiable at all points, similar to $f_1$ and $f_2$. We reconstruct these two functions using two sets of MLP and KAN networks that have comparable parameters but are trained with different sample sizes. The outcomes are displayed in Figure 1. It can be observed that for this category of functions, KAN outperforms MLP.

Table 1: Several Types of Functions and Their Examples

| Regular | Smooth | |
|---|---|---|
| | $f_1(x) = x^2$ | $f_2(x) = e^x$ |
| **Irregular** | Continuous everywhere except at points of non-differentiability | |
| | $f_3(x) = |x|$ | $f_4(x) = 1 - \sqrt{|x|}$ |
| | Jump | |
| | $f_5(x) = \begin{cases} 1, & |x| < 0.5 \\ 0, & \text{other} \end{cases}$ | $f_6(x) = \begin{cases} 1 - 4x^2, & |x| < 0.5 \\ 1, & \text{other} \end{cases}$ |
| | Singular | |
| | $f_7(x) = \frac{1}{x}$ | $f_8(x) = \frac{1}{1-x^2} - 1$ |
| | Coherent oscillation | |
| | $f_9(x) = \cos(\frac{1}{x})$ | $f_{10}(x) = \cos(\frac{2\pi}{1-x^2})$ |
| **Noisy** | Noisy | |
| | $y = x + n(x)$, where $n(x)$ denotes additive noise. | |

Table 2: The number of parameters for each KAN and MLP network.

| MLP | | KAN | | | |
|---|---|---|---|---|---|
| width | parameter | width | grid | k | number of parameters |
| [1,39,1] | 118 | [1,5,1] | 3 | 3 | 120 |
| [1,79,1] | 238 | [1,10,1] | 3 | 3 | 240 |

Table 3: Time consumption of L-BFGS and Adam optimizers in fitting functions $f_7$ and $f_8$ using MLP and KAN

| Function | Network | Optimizer | Time(s) |
|----------|---------|-----------|---------|
| $f_7$ | MLP | L-BFGS | 8.3069 |
| $f_7$ | MLP | Adam | 4.3064 |
| $f_7$ | KAN | L-BFGS | 588.8074 |
| $f_7$ | KAN | Adam | 38.4595 |
| $f_8$ | MLP | L-BFGS | 8.6801 |
| $f_8$ | MLP | Adam | 4.8102 |
| $f_8$ | KAN | L-BFGS | 359.7498 |
| $f_8$ | KAN | Adam | 39.4296 |

Table 4: Time consumption of L-BFGS and Adam optimizers in fitting functions $f_9$ and $f_{10}$ using MLP and KAN

| Function | Network | Optimizer | Time(s) |
|----------|---------|-----------|---------|
| $f_9$ | MLP | L-BFGS | 8.4784 |
| $f_9$ | MLP | Adam | 4.5564 |
| $f_9$ | KAN | L-BFGS | 237.6449 |
| $f_9$ | KAN | Adam | 38.9890 |
| $f_{10}$ | MLP | L-BFGS | 5.8473 |
| $f_{10}$ | MLP | Adam | 4.8208 |
| $f_{10}$ | KAN | L-BFGS | 347.3375 |
| $f_{10}$ | KAN | Adam | 38.6431 |

## 3.2 CONTINUOUS FUNCTIONS WITH POINTS WHERE DERIVATIVES DO NOT EXIST

The functions $f_3$ and $f_4$ serve as prime examples of this category. They maintain continuity at all points, while they are non-differentiable at $x = 0$.

The outcomes are illustrated in Figure 2. For these particular functions, the KAN's performance is worse than the MLP's. Despite the MLP's slower convergence, it eventually reaches a lower test loss. Additionally, it can be noted that amplifying the training sample size marginally enhances the performance of both networks. However, in the vicinity of the non-differentiable point, the MLP shows more significant improvement than the KAN. More visually, it is evident that the fitting performance of MLP and KAN around $x = 0$ (the non-differentiable point) is approximately the same. Yet, with a larger training sample size, the MLP demonstrates superior fitting performance near $x = 0$ compared to the KAN.

## 3.3 FUNCTIONS WITH JUMP

The examples of this category include $f_5$ and $f_6$. These functions have jump discontinuities at $x = \pm 0.5$, where the function values abruptly change between $0$ and $1$. The experimental outcomes for these functions are depicted in Figure 3. Results show that the MLP outperforms the KAN. Moreover, expanding the training dataset size can enhance both networks' performance to a certain extent. Nevertheless, KAN consistently fails to match the performance of MLP.

## 3.4 FUNCTIONS WITH SINGULARITIES

Functions possessing singularities display distinct behaviors, marked by a rapid change rate as they near these points, with their first derivative tending towards infinity at the singularity. Additionally, for any chosen continuous interval that omits these singularities, the functions remain continuous and differentiable across the interval.

To avoid division by zero and guarantee clear fitting results, the ranges of the functions $f_7$ and $f_8$ are limited to $[0.001, 1]$ and $[-0.999, 0.999]$, respectively. We examined the effects of the training sample size, the number of Epochs, and the selection of optimizer on the fitting performance. As illustrated in Figure 4, simply enlarging the sample size by itself does not substantially enhance the performance when recovering $f_7$ and $f_8$.

Pykan offers two optional optimizers: Adam and L-BFGS. As depicted in Figure 5, L-BFGS achieves faster convergence, whereas networks with Adam converges to a lower test loss. However, it is crucial to recognize that with a fixed learning rate, the improvement from this strategy is naturally constrained. As illustrated in Table 3, for an identical number of epochs, the training duration of the network with the L-BFGS optimizer is frequently several times greater than with the Adam optimizer.

Drawing from earlier experiments and findings, the fitting tests will utilize the Adam optimizer set with a learning rate of 0.01. As shown in Figure 6, KAN outperformed MLP in terms of fitting functions with singularities at the same number of epochs.

### 3.5 FUNCTIONS WITH COHERENT OSCILLATIONS

A unique type of functional singularity, labeled as 'coherent oscillatory singularity,' is exemplified by functions $f_9$ and $f_10$. These functions display 'unreachable points' (e.g., $x = 0$ for $f_9$), where as the function approaches these points, its values oscillate increasingly rapidly, intersecting the x-axis infinitely often.

In the experimental phase, taking a similar approach as described in section D. As shown in Figure 4 and 7, an increase in the sampling rate did not markedly enhance fitting accuracy. Particularly, within the KAN network framework, the optimizer L-BFGS outperformed Adam for function $f_9$, while for function $f_{10}$, Adam showed superior results. On the other hand, when fitting both functions with an MLP, Adam consistently performed better than L-BFGS.

In a similar manner, Table 4 demonstrates that employing the L-BFGS optimizer during the fitting process usually resulted in an additional increase in computational time. Figure 8 demonstrates that KAN consistently surpasses MLP when comparing performance over the same number of epochs.

## 4 COMPARISON ON NOISY FUNCTIONS

In the following, we discuss the roles of noisy functions. We introduce noise to functions previously discussed and proceed to evaluate the performance of MLP and KAN. According to the conclusions drawn in the preceding section, we will classify the functions into three categories: regular functions, functions with localized irregularities, and functions with severe discontinuities.

### 4.1 ADDING NOISE TO REGULAR FUNCTIONS

We introduce noise to functions exhibiting strong regularity, and subsequently fit these noisy data using KAN and MLP. The experimental outcomes are depicted in Figure 9. Our observations indicate that KAN achieves a lower test loss with low noise levels but performs worse under high noise conditions. When comparing the function fitting effect, the conclusion remains consistent: MLP shows better performance with minor noise interference, but KAN rapidly outperforms MLP as the training sample size increases.

### 4.2 ADDING NOISE TO FUNCTIONS WITH LOCALIZED IRREGULARITIES

Noise is subsequently added to $f_3$, $f_4$, $f_5$, and $f_6$. The experimental findings are shown in Figure 10. For $f_3$ and $f_4$, the network can still capture some of the irregular features with a larger training sample. However, for $f_5$ and $f_6$, both KAN and MLP perform poorly. The networks still have difficulty identifying the jump discontinuities, even with an increased sample size.

### 4.3 ADDING NOISE TO FUNCTIONS WITH SEVERE DISCONTINUITIES

Figure 11 shows that KAN's performance surpasses that of MLP when noise is added to functions with singularities or coherent oscillation. Interestingly, from the perspective of test loss, the impact of noise on fitting such functions is minimal. This phenomenon highlights the ineffectiveness of strategies relying solely on increased sampling rates in such instances.

## 5 CONCLUSION

In this study, we evaluate the effectiveness of KAN and MLP in approximating irregular or noisy functions. Our analysis concentrates on two main factors: the relative performance of KAN and MLP in fitting functions with different types according to regularity, and their ability to handle noise during the fitting process.

Firstly, as identified in (Shen et al., 2024) and additionally explored in this study, raising the sampling rate is a potent method to enhance the fitting performance of functions $f_1 - f_6$. Particularly, this strategy shows greater advantages when handling noisy data versus clean data. Nevertheless, the improvement in the fitting accuracy for functions with low regularity ($f_7 - f_{10}$) is minimal, irrespective of the presence of noise.

Secondly, we also compared the fitting performance under varying Epochs from two distinct perspectives: convergence speed and stabilized test loss. KAN exhibits a faster convergence rate than MLP across all tested functions. However, MLP outperforms KAN on test functions $f_3 - f_6$ on stabilized test loss.

Thirdly, via experimental analysis (fitting $f_7 - f_{10}$), it was observed that Adam exceeded L-BFGS in performance for both networks in every instance, except for function $f_9$. Notably, when fitting function $f_9$ with the KAN, L-BFGS demonstrated better results than Adam.

At last, when dealing with noisy functions, KAN exhibits superior performance over MLP for regular functions or functions with severe discontinuities. Conversely, for functions with localized irregularities, MLP outperforms KAN.

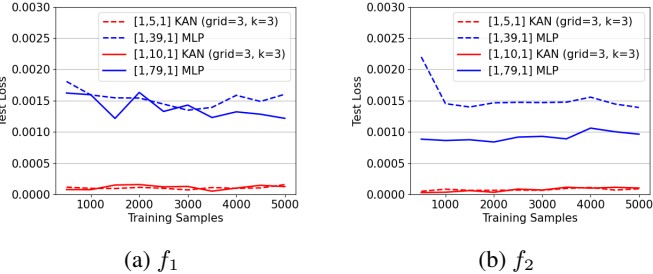

(a) $f_1$       (b) $f_2$

Figure 1: Recover $f_1$ and $f_2$ independently using KAN and MLP under different training sample sizes.

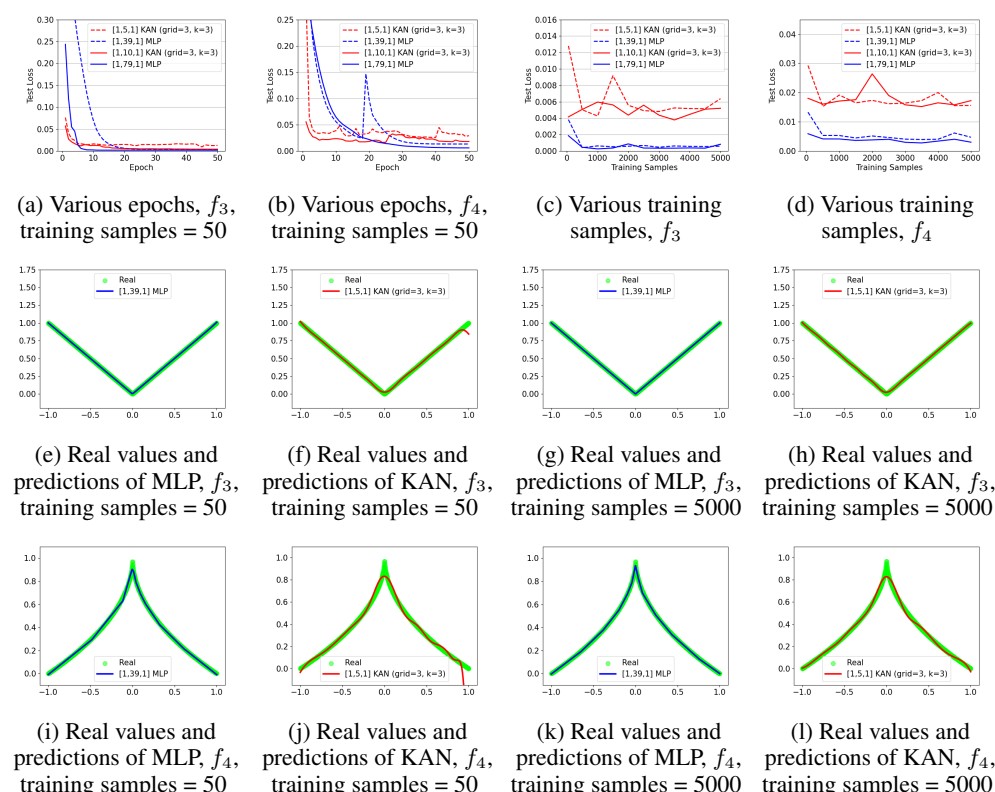

(a) Various epochs, $f_3$, training samples = 50

(b) Various epochs, $f_4$, training samples = 50

(c) Various training samples, $f_3$

(d) Various training samples, $f_4$

(e) Real values and predictions of MLP, $f_3$, training samples = 50

(f) Real values and predictions of KAN, $f_3$, training samples = 50

(g) Real values and predictions of MLP, $f_3$, training samples = 5000

(h) Real values and predictions of KAN, $f_3$, training samples = 5000

(i) Real values and predictions of MLP, $f_4$, training samples = 50

(j) Real values and predictions of KAN, $f_4$, training samples = 50

(k) Real values and predictions of MLP, $f_4$, training samples = 5000

(l) Real values and predictions of KAN, $f_4$, training samples = 5000

Figure 2: Recover $f_3$ and $f_4$ independently using KAN and MLP.

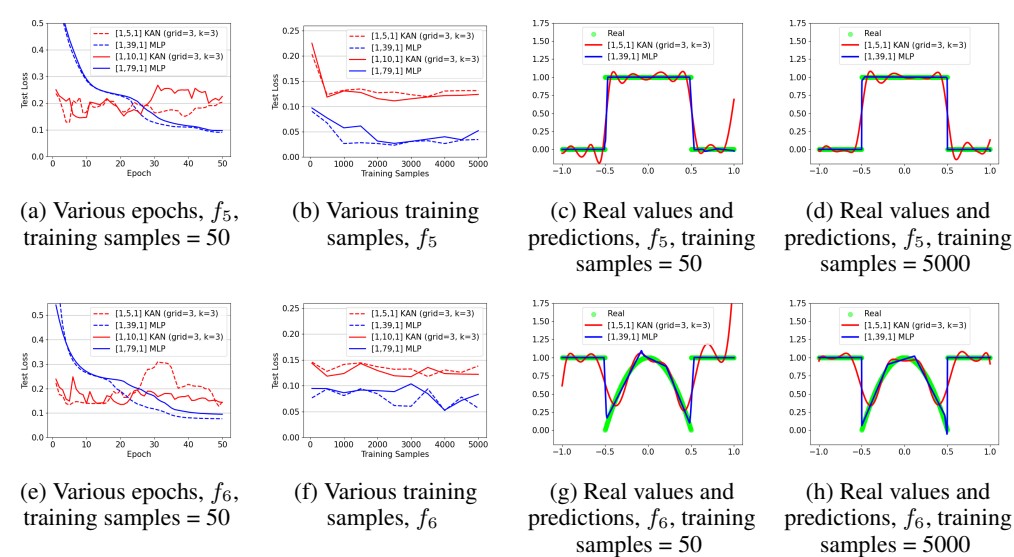

(a) Various epochs, $f_5$, training samples = 50

(b) Various training samples, $f_5$

(c) Real values and predictions, $f_5$, training samples = 50

(d) Real values and predictions, $f_5$, training samples = 5000

(e) Various epochs, $f_6$, training samples = 50

(f) Various training samples, $f_6$

(g) Real values and predictions, $f_6$, training samples = 50

(h) Real values and predictions, $f_6$, training samples = 5000

Figure 3: Recover $f_5$ and $f_6$ independently using KAN and MLP.

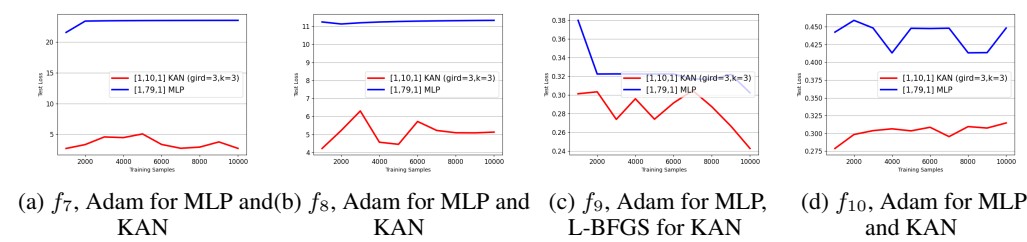

(a) $f_7$, Adam for MLP and KAN

(b) $f_8$, Adam for MLP and KAN

(c) $f_9$, Adam for MLP, L-BFGS for KAN

(d) $f_{10}$, Adam for MLP and KAN

Figure 4: Recover $f_7$, $f_8$, $f_9$ and $f_{10}$ independently using KAN and MLP with optimizer Adam or L-BFGS, 2000 Epochs

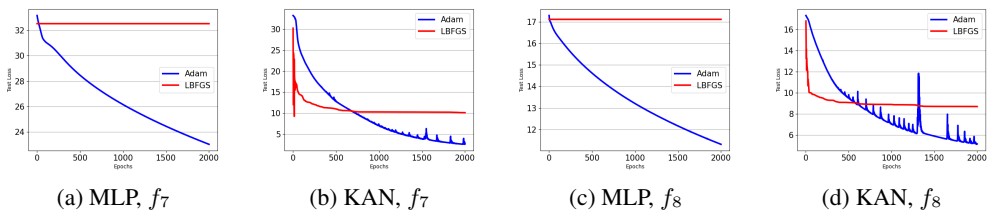

(a) MLP, $f_7$

(b) KAN, $f_7$

(c) MLP, $f_8$

(d) KAN, $f_8$

Figure 5: The variation of test loss with the increasing number of epochs when recovering $f_7$ and $f_8$ independently using [1,10,1] KAN (grid=3,k=3) and [1,79,1] MLP with optimizer L-BFGS and Adam, learning rate=0.01

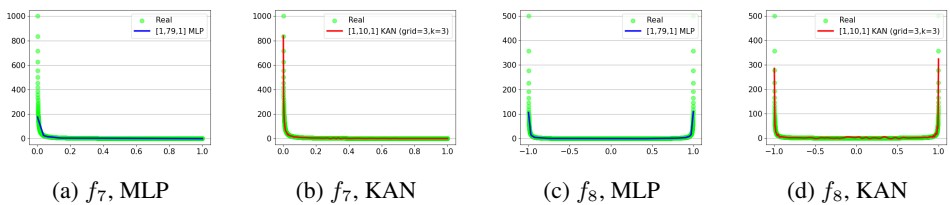

(a) $f_7$, MLP

(b) $f_7$, KAN

(c) $f_8$, MLP

(d) $f_8$, KAN

Figure 6: Recover $f_7$, $(x \in [0.001, 1])$ and $f_8$, $(x \in [-0.999, 0.999])$ independently using [1,10,1] KAN (grid=3, k=3) and [1,79,1] MLP both with optimizer Adam, same or different epochs

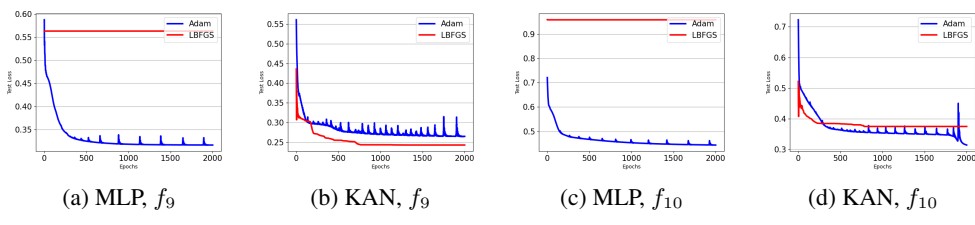

(a) MLP, $f_9$  (b) KAN, $f_9$  (c) MLP, $f_{10}$  (d) KAN, $f_{10}$

Figure 7: The variation of test loss with the increasing number of epochs when recovering $f_9$ and $f_{10}$ independently using [1,10,1] KAN (grid=3,k=3) and [1,79,1] MLP with optimizer L-BFGS and Adam, learning rate=0.01

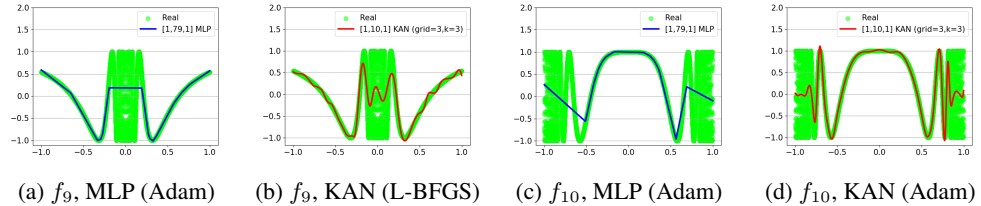

(a) $f_9$, MLP (Adam)  (b) $f_9$, KAN (L-BFGS)  (c) $f_{10}$, MLP (Adam)  (d) $f_{10}$, KAN (Adam)

Figure 8: Recover $f_9, (x \in [-0.999, 0.999])$ and $f_{10}, (x \in [-0.999, 0.999])$ independently using [1,10,1] KAN (grid=3,k=3) and [1,79,1]MLP with same or different optimizers and epochs

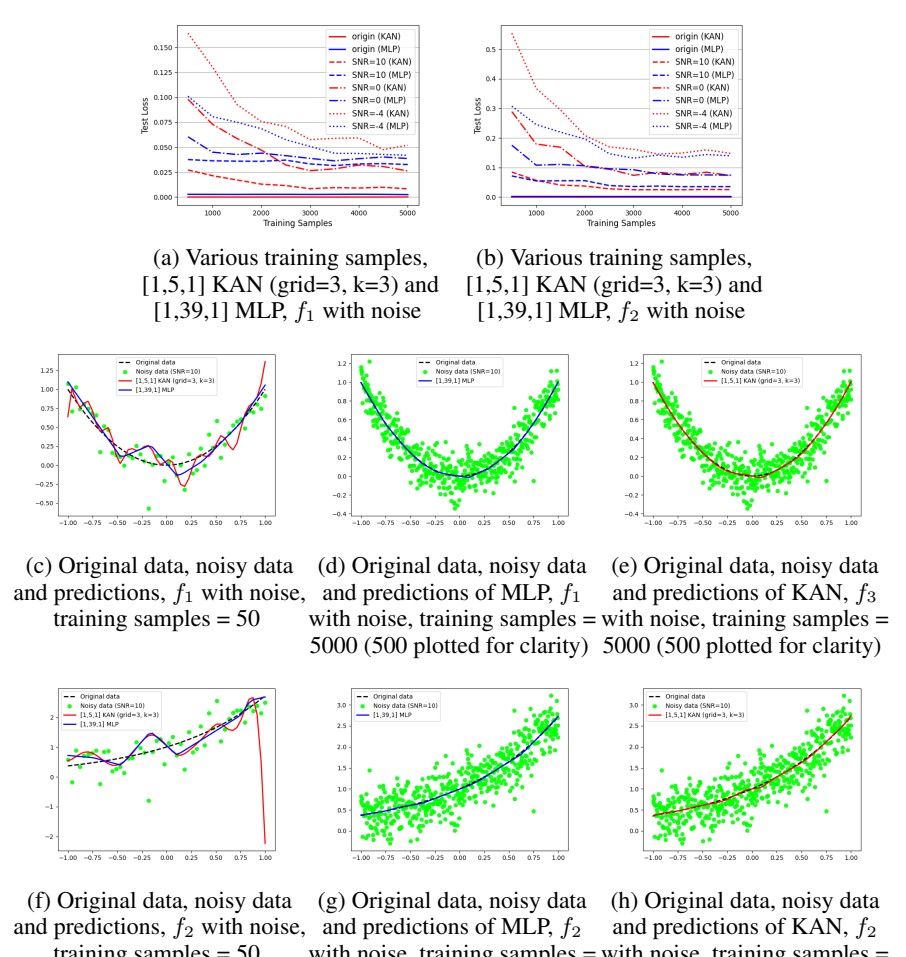

(a) Various training samples, [1,5,1] KAN (grid=3, k=3) and [1,39,1] MLP, $f_1$ with noise

(b) Various training samples, [1,5,1] KAN (grid=3, k=3) and [1,39,1] MLP, $f_2$ with noise

(c) Original data, noisy data and predictions, $f_1$ with noise, training samples = 50

(d) Original data, noisy data and predictions of MLP, $f_1$ with noise, training samples = 5000 (500 plotted for clarity)

(e) Original data, noisy data and predictions of KAN, $f_3$ with noise, training samples = 5000 (500 plotted for clarity)

(f) Original data, noisy data and predictions, $f_2$ with noise, training samples = 50

(g) Original data, noisy data and predictions of MLP, $f_2$ with noise, training samples = 5000 (500 plotted for clarity)

(h) Original data, noisy data and predictions of KAN, $f_2$ with noise, training samples = 5000 (500 plotted for clarity)

Figure 9: Recover $f_1$ and $f_2$ with noise independently using KAN and MLP.

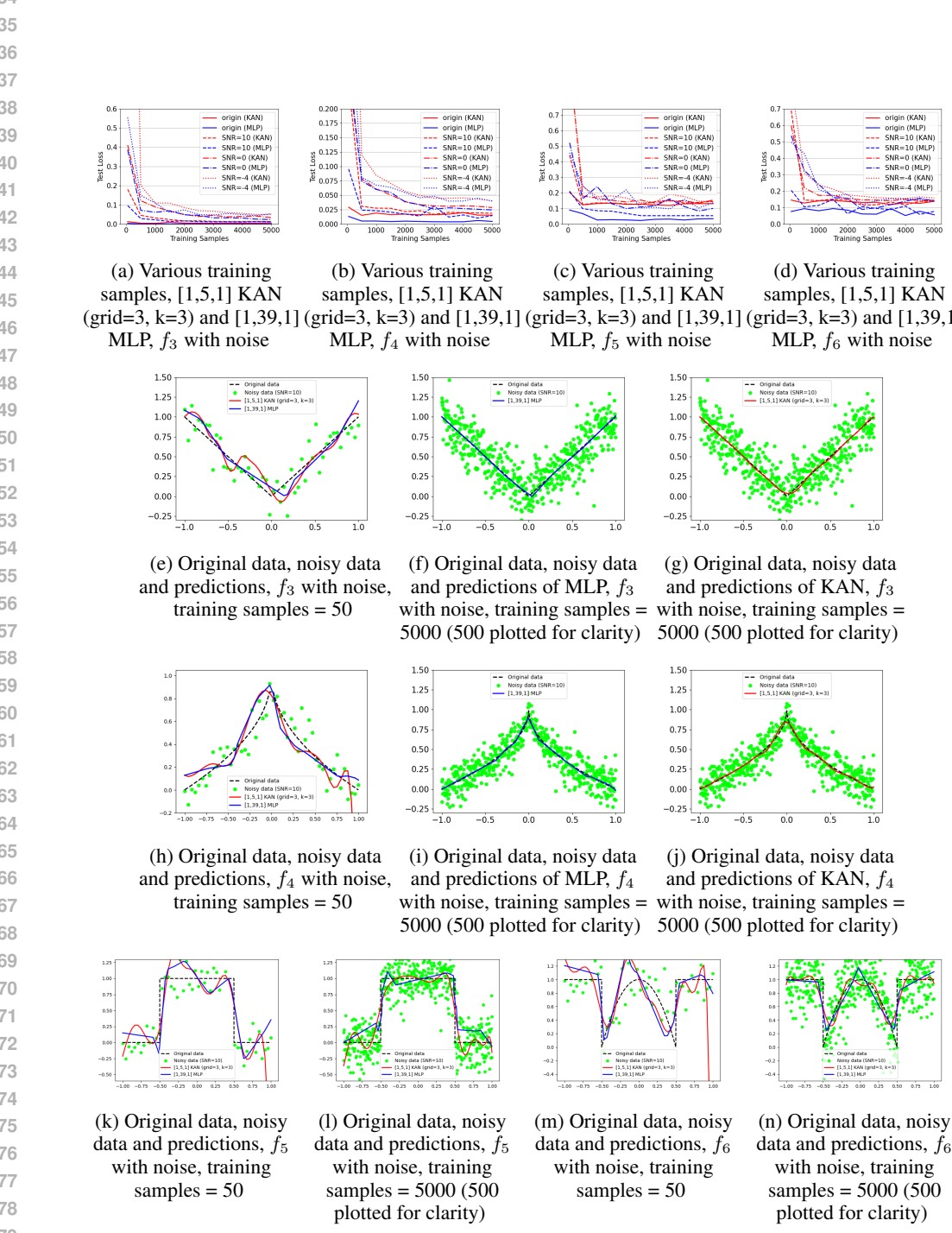

(a) Various training samples, [1,5,1] KAN (grid=3, k=3) and [1,39,1] MLP, $f_3$ with noise

(b) Various training samples, [1,5,1] KAN (grid=3, k=3) and [1,39,1] MLP, $f_4$ with noise

(c) Various training samples, [1,5,1] KAN (grid=3, k=3) and [1,39,1] MLP, $f_5$ with noise

(d) Various training samples, [1,5,1] KAN (grid=3, k=3) and [1,39,1] MLP, $f_6$ with noise

(e) Original data, noisy data and predictions, $f_3$ with noise, training samples = 50

(f) Original data, noisy data and predictions of MLP, $f_3$ with noise, training samples = 5000 (500 plotted for clarity)

(g) Original data, noisy data and predictions of KAN, $f_3$ with noise, training samples = 5000 (500 plotted for clarity)

(h) Original data, noisy data and predictions, $f_4$ with noise, training samples = 50

(i) Original data, noisy data and predictions of MLP, $f_4$ with noise, training samples = 5000 (500 plotted for clarity)

(j) Original data, noisy data and predictions of KAN, $f_4$ with noise, training samples = 5000 (500 plotted for clarity)

(k) Original data, noisy data and predictions, $f_5$ with noise, training samples = 50

(l) Original data, noisy data and predictions, $f_5$ with noise, training samples = 5000 (500 plotted for clarity)

(m) Original data, noisy data and predictions, $f_6$ with noise, training samples = 50

(n) Original data, noisy data and predictions, $f_6$ with noise, training samples = 5000 (500 plotted for clarity)

Figure 10: Recover $f_3$, $f_4$, $f_5$ and $f_6$ with noise independently using KAN and MLP.

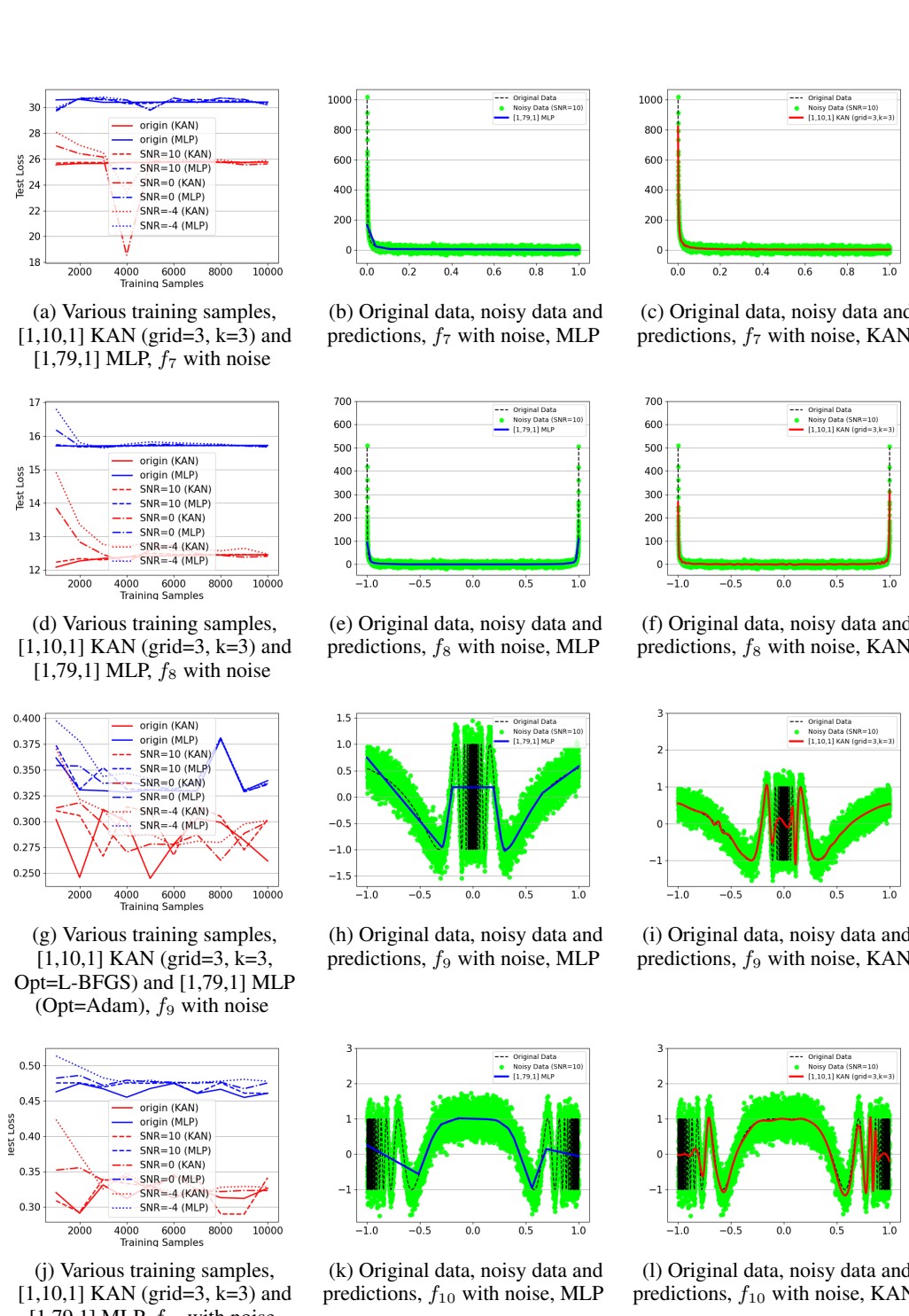

(a) Various training samples, [1,10,1] KAN (grid=3, k=3) and [1,79,1] MLP, $f_7$ with noise

(b) Original data, noisy data and predictions, $f_7$ with noise, MLP

(c) Original data, noisy data and predictions, $f_7$ with noise, KAN

(d) Various training samples, [1,10,1] KAN (grid=3, k=3) and [1,79,1] MLP, $f_8$ with noise

(e) Original data, noisy data and predictions, $f_8$ with noise, MLP

(f) Original data, noisy data and predictions, $f_8$ with noise, KAN

(g) Various training samples, [1,10,1] KAN (grid=3, k=3, Opt=L-BFGS) and [1,79,1] MLP (Opt=Adam), $f_9$ with noise

(h) Original data, noisy data and predictions, $f_9$ with noise, MLP

(i) Original data, noisy data and predictions, $f_9$ with noise, KAN

(j) Various training samples, [1,10,1] KAN (grid=3, k=3) and [1,79,1] MLP, $f_{10}$ with noise

(k) Original data, noisy data and predictions, $f_{10}$ with noise, MLP

(l) Original data, noisy data and predictions, $f_{10}$ with noise, KAN

Figure 11: Recover $f_7$, $f_8$, $f_9$ and $f_{10}$ with noise independently using KAN and MLP.

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
