# OpenReview forum: "KAN versus MLP on Irregular or Noisy Functions"
_ICLR.cc/2025/Conference — Submitted to ICLR 2025_

### Official Review · Reviewer_FMgf · 2024-10-27

**Soundness:** 2
**Presentation:** 2
**Contribution:** 1
**Rating:** 3
**Confidence:** 4

**Summary:**

In this empirical study, the authors compare the performance of two ad-hoc versions of KAN and MLPs, both with identical parameter counts, in learning ten single-dimensional real functions, with and without added noise.

**Strengths:**

The work has considered different classes of functions with some common irregularities. The empirical comparison is sound, and extensive plots provided let's the reader to compare the performance of tested KAN and MLPs in each case.

**Weaknesses:**

The comparison primarily centers on the resulting test accuracy curves; however, it lacks the necessary theoretical justification and fundamental analysis to substantiate the findings.

While the authors have structured the text well, the plots are somewhat cluttered and could be presented more effectively for better clarity.

Overall, the work appears basic and does not demonstrate the level of novelty typically expected from submissions to ICLR.

**Questions:**

If authors can provide some theoretical insights backing the observed empirical results in all or some of the function classes tested, it would make the work more promising and considerable for this conference.

**Details Of Ethics Concerns:**

The authors cite a paper with "our work" in the first page, fourth paragraph of Introduction. This seems to be an instance of violation of anonymity of the authors in the double-blind review process.

---

> ### Author Response · Authors · 2024-11-20
>
> Thank you for your valuable time and insightful suggestions. We have carefully considered and thoroughly discussed your suggestions. Below, we provide our detailed responses to your comments.
> > Q1: It lacks the necessary theoretical justification and fundamental analysis to substantiate the findings.
>
> A1: This paper is experimental rather than theoretical, aiming to compare the performance of KAN and MLP in fitting functions with poor regularity through experiments.
>
> > Q2: The authors cite a paper with "our work" in the first page, fourth paragraph of Introduction. This seems to be an instance of violation of anonymity of the authors in the double-blind review process.
>
> A2: Thank you for your careful reading. We deeply apologize for this, it was indeed our oversight. We have corrected this error in the new version.

---

### Official Review · Reviewer_nCqP · 2024-10-29

**Soundness:** 3
**Presentation:** 3
**Contribution:** 1
**Rating:** 3
**Confidence:** 4

**Summary:**

The paper conducts a comparative analysis of experiments between MLP and KANs, discussing the outcomes. It challenges the assumption that KANs consistently outperform MLP in modeling mathematical equations, particularly with irregular functions. The experiments involve applying MLP and KAN to various functions—regular, non-differentiable, discontinuous, singular, and coherent oscillation, with and without noise. These functions are single input and single output. Variations include different training sample sizes, iteration counts, and optimizers. The findings demonstrate that KANs do not always surpass MLP.

While this paper serves as a great exploration to KANs and does establish that KANs are not invariably superior to MLP, it falls short by only providing experimental evidence without introducing new theoretical insights or network structures, thus lacking substantial academic contribution.

**Strengths:**

The structure of the paper is clear and well-organized.
The experimental results are clearly presented.
The experiments validate that KANs are not consistently superior to MLP.

**Weaknesses:**

The experiments could be designed more targeted. For instance, in the experiments for non-differentiability, both functions feature only a single non-differentiable point. A comparison between functions with single versus multiple non-differentiable points would be more insightful, given the focus on the impact of these points.

The discussion lacks depth. Given the simplicity of both the functions and network structures used, there is potential for a more detailed examination of how parameters are trained and the reasons behind specific outcomes.

The discussion section does not yield any intriguing or unexpected conclusions, nor does it propose any novel theories or structures.

**Questions:**

Given that the structures of both MLP and KANs are well-known, a deeper analysis of their capabilities and limitations in the related work section would be beneficial. More thorough research could uncover more significant findings. For instance, some limitations of KANs identified in the paper are not due to the Kolmogorov-Arnold Theorem but rather due to B-spline, a critical component in KANs that is not discussed in the paper at all.

---

> ### Author Response · Authors · 2024-11-20
>
> Thank you for your valuable time and insightful suggestions. We have carefully considered and thoroughly discussed your suggestions. Below, we provide our detailed responses to your comments.
> > Q1: A comparison between functions with single versus multiple non-differentiable points would be more insightful, given the focus on the impact of these points.
>
> A1: Thank you for your suggestion. The aim of our paper is to demonstrate the potential issues KAN networks may encounter when handling functions with poor regularity. A detailed exploration of the impact of irregular points on KAN would require significant space and is better suited for a separate paper. Therefore, we chose to use simpler functions to illustrate the phenomenon.
>
> > Q2: The discussion lacks depth. Given the simplicity of both the functions and network structures used, there is potential for a more detailed examination of how parameters are trained and the reasons behind specific outcomes. And the discussion section does not yield any intriguing or unexpected conclusions, nor does it propose any novel theories or structures.
>
> A2: This paper is experimental rather than theoretical, aiming to compare the performance of KAN and MLP in fitting functions with poor regularity through experiments.
>
> > Q3: Some limitations of KANs identified in the paper are not due to the Kolmogorov-Arnold Theorem but rather due to B-spline, a critical component in KANs that is not discussed in the paper at all.
>
> A3: Thank you for your suggestion. This is a very interesting idea. However, to validate this claim, we might need to introduce "KAN" model using other basis functions for discussion, which is beyond the scope of our paper.

---

### Official Review · Reviewer_pbqY · 2024-11-04

**Soundness:** 1
**Presentation:** 2
**Contribution:** 1
**Rating:** 3
**Confidence:** 3

**Summary:**

Authors compare the performance of Kolmogorov-Arnold Networks (KAN) and Multi-Layer Perceptron (MLP) networks on irregular or noisy functions. The author experimentally demonstrated that KAN does not always outperform MLP.

**Strengths:**

- The author compared KAN and MLP on various irregular and noisy functions and experimentally demonstrated in which cases KAN is worse than MLP.

**Weaknesses:**

- The author merely compared KAN and MLP experimentally but did not analyze why KAN or MLP performs poorly in certain situations.

- The author experimentally demonstrated that KAN is sometimes inferior to MLP. It would be better to propose a new, improved KAN model to address this.

- There are no experiments on high-dimensional functions. In one dimension, both KAN and MLP are likely to approximate well to some extent, but more experiments are needed to explore how they perform in high-dimensional spaces with irregular points.

- If the experiments are conducted only on univariate functions, many models besides MLP can be compared with KAN. It would be beneficial to include other models commonly used in machine learning in the experiments.

**Questions:**

I do not have a complete understanding of KAN, but I think KAN appears to be a generalization of projection pursuit regression. Is this correct?

---

> ### Author Response · Authors · 2024-11-20
>
> Thank you for your valuable time and insightful suggestions. We have carefully considered and thoroughly discussed your suggestions. Below, we provide our detailed responses to your comments.
> > Q1: The author merely compared KAN and MLP experimentally but did not analyze why KAN or MLP performs poorly in certain situations. And it would be better to propose a new, improved KAN model to address this.
>
> A1: Thank you for your suggestion. This paper is experimental rather than theoretical, aiming to compare the performance of KAN and MLP in fitting functions with poor regularity through experiments.
>
> > Q2: There are no experiments on high-dimensional functions.
>
> A2: We aim to focus on poor regularity rather than the types of functions. And as demonstrated in our paper, even with simple functions, KAN shows certain problems.
>
> > Q3: It would be beneficial to include other models commonly used in machine learning in the experiments.
>
> A3: The primary neural networks under consideration are MLP and KAN, and when KAN was initially proposed, it was primarily compared to MLP. Therefore, we believe it is unnecessary to include additional models in our study.
>
> > Q4: I do not have a complete understanding of KAN, but I think KAN appears to be a generalization of projection pursuit regression. Is this correct?
>
> A4: KAN and projection pursuit regression (PPR) do share some similarities. Both rely on projecting data into a lower-dimensional space and applying nonlinear functions to these projections for regression. PPR works by finding the best linear projections to reveal the structure of the data, and then fitting nonlinear functions to these projections. In contrast, KAN networks are based on the Kolmogorov-Arnold theorem, representing complex multidimensional functions as a combination of several one-dimensional functions. Therefore, KAN can be seen as a generalization and extension of PPR.

---

### Official Review · Reviewer_idCx · 2024-11-04

**Soundness:** 1
**Presentation:** 1
**Contribution:** 1
**Rating:** 1
**Confidence:** 4

**Summary:**

This work empirically compares Kolmogorov-Arnold Networks with Multi-Layer Perceptron on
learning irregular or noisy functions. The experiment results show that KAN do not always perform
the best.

**Strengths:**

Experiment codes are provided for reproducibility.

Do provide some insight on what KAN may be good at modeling.

**Weaknesses:**

The finding is purely empirical.

The paper does not clearly state the experiment setting in the main text.

The experiment does not provide conclusive results.

The experiment only tries to fit relatively simple functions. The result may not be relevant to real-world problems.

**Questions:**

It is possible to include more challenging problems for comparison? It is well established that MLP can model fairly complicated functions.

---

> ### Author Response · Authors · 2024-11-20
>
> Thank you for your valuable time and insightful suggestions. We have carefully considered and thoroughly discussed your suggestions. Below, we provide our detailed responses to your comments.
> > Q1: The finding is purely empirical.
>
> A1: This paper is experimental rather than theoretical, aiming to compare the performance of KAN and MLP in fitting functions with poor regularity through experiments.
>
> > Q2: The paper does not clearly state the experiment setting in the main text.
>
> A2: We did not include the experimental setting in the main text but provided them in the figure captions and tables.
>
> > Q3: The experiment does not provide conclusive results.
>
> A3: Our paper does not aim to prove that KAN is inherently better or worse than MLP. Instead, it seeks to highlight potential issues KAN may encounter when fitting functions with poor regularity. Besides, as demonstrated in the paper, KAN's performance varies across different types of functions. Therefore, we opted to provide conclusions separately for each type of function.
>
> > Q4: The experiment only tries to fit relatively simple functions. The result may not be relevant to real-world problems. Is it possible to include more challenging problems for comparison? It is well established that MLP can model fairly complicated functions.
>
> A4: We aim to focus on poor regularity rather than the types of functions or tasks. As demonstrated in our paper, even with simple functions, KAN shows certain problems.

---

### Meta-Review · Area_Chair_cTKc · 2024-12-20

**Metareview:**

The paper compares the performance of Kolmogorov-Arnold Networks and Multi-Layer Perceptrons in approximating various types of functions, including irregular and noisy ones. Experiments demonstrate that while KANs often outperform MLPs for certain regular or singular functions, MLPs show better performance for functions with localized irregularities or jumps.

The paper is motivated by an interesting question but lacks sufficient evidence and deep discussion. It also offers no new interpretations, theoretical implications, or methodological suggestions for improving the problem. The reviewers agree that much can still be done with this paper.

**Additional Comments On Reviewer Discussion:**

Despite differences in detail, the reviewers were generally in agreement. The paper is limited to basic experimental results, and the experiments alone leave much room for more complex and realistic settings. There is also room for deeper consideration of theory, methodology, and discussion.

---

### Decision · Program_Chairs · 2025-01-22

Reject